The Company of
Biologists

# Insights into the thermal ecology, physiology, and behavior of a threatened ectothermic specialist from a warming and drying ecoregion

Brian R. Blais[1,*], Maria Vittoria Mazzamuto[2,3] and John L. Koprowski[1,2]

## ABSTRACT

Increased heat and drought from Anthropogenic climate change will challenge the adaptive capacity of species, underscoring the need to understand thermal ecology – how organisms behaviorally and physiologically respond to temperature. We used noninvasive infrared thermography (IRT) to examine the thermal ecology of threatened narrow-headed gartersnakes (*Thamnophis rufipunctatus*) in a conservation breeding program at the Arizona Center of Nature Conservation/Phoenix Zoo. From 718 microhabitat and 124 individual measurements, hierarchical models identified extrinsic and intrinsic factors influencing microhabitat usage, body temperature ($T_b$), and behavior. Gartersnakes exhibited regional heterothermy, with tails cooler than head and trunk segments. The $T_b$ of *T. rufipunctatus* was shaped by perch temperature, perch-air temperature, and whether snakes were visibly exposed or hidden. We documented microhabitat aggregations ($\geq 2$ gartersnakes) in ca. 40% of observations, which was best predicted by $T_b$. *Thamnophis rufipunctatus* appeared to favor cavity-bearing microhabitats, consistent with wild populations. This first application of IRT to snakes in semi-natural environments, and for *T. rufipunctatus* specifically, provides novel insights to guide more effective field surveillance and conservation management, while demonstrating the broader value of IRT and collaborative *ex situ* studies for wildlife conservation.

KEY WORDS: Body temperature, Conservation, Endangered species, Infrared thermography, Reptiles, Zoos

## INTRODUCTION

Contemporary rates of rising temperature and prolonged drought under Anthropogenic climate change will challenge the pace at which species can adaptively respond (Bennett et al., 2021; Hof et al., 2011; Mitchell et al., 2018; Radchuk et al., 2019; Visser, 2008), which presents a major threat to Earth's biodiversity (Bellard et al., 2012; Pacifici et al., 2015; Pecl et al., 2017). The impacts of rapid onset changes on bio-available water, metabolism, and reproduction in amphibians and reptiles, for example, can be severe or irreversible (Bickford et al. 2010, Davis et al. 2015, Mitchell et al. 2016). The thermal tolerance and responses to environmental stimuli of a species first require an understanding of its thermal ecology – an organism's behavioral and physiological response to temperature (Bogert, 1959; Camacho et al., 2018; Refsnider et al., 2019). Endothermic animals, such as birds and mammals, generate their own heat through internal metabolism and need mechanisms to conserve it; to maintain body temperature ($T_b$), heat produced in the core is distributed to the periphery (Hill et al., 2016; Scholander et al., 1950). Ectothermic reptiles rely on an interplay of environmental stimuli (e.g. conduction, convection, radiation), metabolic physiology, morphology, and behavior to thermoregulate $T_b$ to optimal levels to perform natural activities (Christian et al., 2016; Cowles and Bogert, 1944; Cox et al., 2022; Goulet et al., 2017; Kearney et al., 2009).

Many reptiles select microhabitats (e.g. burrows/refuges, basking/ perch structures) to thermoregulate as well as perform regular natural behaviors such as foraging, escaping predation, gestating, and aestivation/overwintering among others (Dubiner et al., 2024; Goulet et al., 2017; Harvey and Weatherhead, 2010; Sannolo et al., 2019; Scheffers et al., 2014). Ambient conditions alone may not capture thermal heterogeneity within and among the microhabitat landscape; i.e. microclimate (Campobello et al., 2017; Kearney et al., 2009; Scheffers et al., 2016; Woods et al., 2015). Infrared thermography (IRT; Tattersall, 2016; McCafferty et al., 2021) tools can noninvasively and simultaneously capture the thermal properties of focal species and the microhabitat surfaces being occupied, which can help infer site selection (Blais et al., 2023a; Goller et al., 2014; Krochmal and Bakken, 2003; Signore et al., 2020; Taylor et al., 2021). Ecological insights garnered from IRT include various anatomical, physiological, and behavioral adaptations to thermal environments (Mazzamuto et al., 2023; McCafferty et al., 2021; Mitchell and Clarke, 2019; Monge et al., 2025). Understanding environmental influences of favored microhabitats can guide more effective conservation actions, such as species reintroduction site planning or habitat restoration initiatives (Huey et al., 1989; Luna and Font, 2013).

Assessing thermal physiology in wild systems allow for natural behaviors but complexities and stressors exist (e.g. nutrient requirements, threat of predation), whereas (simple) *ex situ* arenas offer stronger experimental control but could influence behavior or other aspects of thermal ecology, including diel/seasonal changes, sex, gravidity, and age-class, especially for at-risk species (Tattersall, 2016; Taylor et al., 2021). *Ex situ* (e.g. zoos, aquariums) managed semi-natural environments and enrichment strategies that facilitate innate behavior and needs of focal species may provide a compromise through informational feedback (Blais et al., 2023b; Burghardt, 2013; Chiszar et al., 1993; Reading et al., 2013). Zoo populations can present opportunities to closely monitor and gain insight into many natural parameters (e.g. behavior, life history traits, physiology) of a

[1]School of Natural Resources and the Environment, University of Arizona, ENR2,1064 E. Lowell St., Tucson, AZ 85721, USA. [2]Haub School of Environment & Natural Resources, University of Wyoming, 201 Bim Kendall House, 804 E Fremont St., Laramie, WY 82072, USA. [3]Department of Life Sciences and Systems Biology, University of Turin, 10123 Turin, Italy.

*Author for correspondence (bblais@arizona.edu)

B.R.B., 0000-0001-9633-5259; M.V.M., 0000-0002-4728-0527; J.L.K., 0000-0003-1406-9853

focal species that may otherwise be data deficient or onerous to track *in situ* (Blais et al., 2023b; Minteer et al., 2018; Pritchard et al., 2011; Spooner et al., 2023).

The narrow-headed gartersnake, *Thamnophis rufipunctatus* (Cope, in Yarrow, 1875), is a piscivorous specialist endemic to montane, perennial, cool-water riparian systems (700–2430 m asl) in the Gila River Watershed of central Arizona and western New Mexico (Holycross et al., 2020; Rossman et al., 1996; Wood et al., 2018). Limited information exists on the thermal physiology of *T. rufipunctatus* (USFWS, 2014). Body temperature has been shown to strongly relate to air and surface temperature (Fleharty, 1967) but also water temperature (Hibbitts et al., 2009). This species is active in a wide range of air temperature ($T_a$, 11–32°C) and water temperature ($T_w$, 12–22°C; USFWS, 2014); active *T. rufipunctatus* can operate at cooler conditions than several congenerics (Rosen, 1991), including sympatric species (Fleharty, 1967). Range-wide population declines (Holycross et al., 2020; Wood et al., 2025a) culminated in the species being listed as threatened under the Endangered Species Act (USFWS, 2014; Wood et al., 2018). Projected loss of suitable environment across time in a warming and drying region presents additional challenges the species is likely to face (Blais and Koprowski, 2024). In response, the Arizona Center for Nature Conservation/Phoenix Zoo (hereafter ACNC) has managed an *ex situ* conservation breeding program for *T. rufipunctatus* since 2006 (Allard and Wells, 2018). The *ex situ* colony at ACNC has provided opportunities to study and learn about this enigmatic species, including insights into behavior and reproductive biology (Blais et al., 2023b; Wood et al., 2025b).

Herein, we conducted a noninvasive IRT study on the ACNC colony to better understand the thermal ecology and physiology of *T. rufipunctatus*. Our objectives were to: 1) model environmental and temporal factors in snake-used versus available microhabitats in two complex naturalistic microcosms; 2) derive $T_b$ and thermal physiology metrics; 3) analyze how behavior and environment influence $T_b$; and 4) explore relationships of intrinsic and extrinsic factors that may influence aggregation behavior. We expected gartersnakes to select microhabitat types similar to those used in the wild (e.g. refuges with cavities; Holycross et al., 2020; Rosen, 1991) and in relation to thermoregulatory needs. We expected $T_b$ measures to be near or slightly above measurements recorded in the wild due to the controlled environment maintained in the *ex situ* location. We use *ex situ* ecological and physiological data to guide conservation of an imperiled ectotherm and inform broader thermal ecology research. Our methods incentivize IRT use in naturalistic settings and strengthen feedback loops and collaboration across the *ex situ–in situ* spectrum. Understanding how body temperature, behavior, and microclimate interact can reveal thermal preferences and adaptive challenges terrestrial ectotherms face in a warming, drying climate.

## RESULTS
### General results and temporal conditions
We completed 36 total surveys (1–2 per enclosure per sampling date) between 24 May and 27 September 2019 (sampling date interval: 14±0.7 d); we ceased the project in conjunction with planned maintenance by ACNC in preparation for the overwinter brumation period. Morning shift surveys occurred at 10:12 h on average and afternoon shift surveys at 13:04 h on average (shift interval: 178.1±35.2 min); we omitted two afternoon surveys due to logistical restraints/husbandry priorities. The average survey duration was 13.9 (±5.8) min. Ambient conditions varied temporally between morning and afternoon survey shifts for $P_b$ (Wilcoxon test: $P<0.001$; mean change=−2.2 mb), rH ($P<0.001$; mean change=−2.9%), $T_a$ ($P<0.001$; mean change=2.7°C), and $T_w$ ($P<0.001$; mean change=2.1°C;

Fig. S2A,D). Ambient conditions also varied by month (Kruskal–Wallis: all $P<0.001$).

### Microhabitat selection by narrow-headed gartersnakes
We generated a dataset of $n$=718 microhabitat assessments. The thermal heterogeneity of microhabitats varied within and among types ($T_s$: 29.8±4.55°C, $T_s$-$T_a$: 0.4±3.35°C; Table S1, Fig. S3A,B). Although we knew gartersnake total $N$ per enclosure survey, we did not always observe all present individuals (detection rate: 60.1±29.7%) and made no attempt to search beyond surface visuals (i.e. beneath substrates, leafy plants, or underwater). Gartersnakes occupied microhabitat types as follows: subterranean (62.9%), cover (22.4%), water (10.3%), ground (2.3%), and plants (2.0%). Twenty-seven of 42 unique microhabitats remained unoccupied during study observation.

Four models were within two corrected Akaike Information Criterion (AICc) units to explain microhabitat selection (Table S2). Only microhabitat type was identified as an important variable at the ≥0.80 threshold score and was the only factor in the top microhabitat usage model (AICc=312.6, weight=0.188, $R^2$=0.71). After investigating the internal-only microhabitat dataset, there were seven models within two AICc units (Table S2). Again, type was important at ≥0.80 threshold although $T_a$ and (internal) $T_s$ had scores 0.40–0.60. The model that included those three terms yielded some trends (0.05<$P$<0.10) that *T. rufipunctatus* were more likely to use microhabitats with internal structures when ambient $T_a$ increased and microhabitat internal $T_s$ decreased. That is, cover microhabitats provided refuge to cool down.

### Thermal ecology of *T. rufipunctatus*
We amassed $n$=124 assessments for *T. rufipunctatus*. The body temperature dataset included $n$=117 records after omitting seven events that lacked $T_b$ (e.g. indiscernible clustered individuals; snake fled before measurement). From $n$=74 measurement events that had data for all three integuments, there was evidence of regional heterothermy ($F_{144}$=14.2, $η^2$=0.004, $P<0.001$). Gartersnake head (27.2±2.9°C) and trunk segments (27.2±3.0°C) were equivalent, but tail segments were slightly but significantly cooler (26.8±3.0°C; Fig. 1). Effects from exposure behavior ($P$=0.237), i.e. whether snakes were hidden within refuges (e.g. thermoregulatory cooling) or visibly

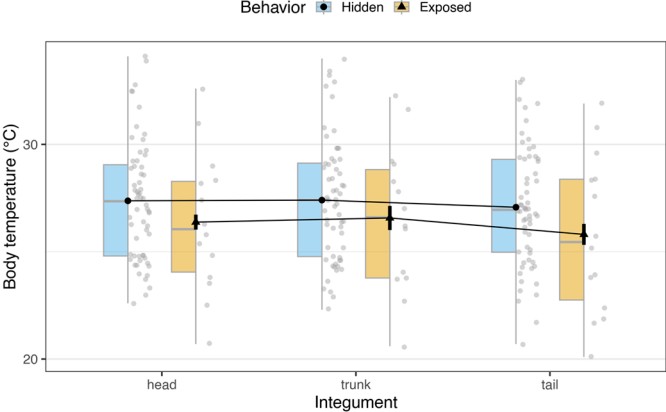

**Fig. 1. Regional heterothermy by integument for *ex situ* colony of *T. rufipunctatus* at the Phoenix Zoo, 2019.** Data are external body temperature values ($n$=74) obtained from infrared thermography for gartersnake head, trunk, and tail (post-cloaca) integuments. Data are partitioned by gartersnake behavior, whether they were observably visible and exposed to elements (e.g. basking) or hidden within refuges (e.g. cooling).

exposed (e.g. surface basking), or interaction effects ($P$=0.142) were not found. We note that assessing individual integuments on analyses hereafter did not alter results.

The population $T_b$ metrics for *T. rufipunctatus* (*n*=117) were 27.2±2.94°C. This central tendency and dispersion varied from populations in older studies ($Z$>3, $P$<0.001; Fleharty, 1967; Rosen, 1991; Hibbitts et al., 2009) but was comparable to a more recent assessment ($Z$=1.7, $P$=0.082; see Jennings and Christman in USFWS, 2014; Table 1). The preferred body temperature was as follows: median (i.e. $T_{set}$)=27.4°C, $T_{set}$ lower bounds=24.6°C, $T_{set}$ upper bounds=29.4°C, $T_{set-range}$=4.8°C (Fig. 2). Thermoregulatory accuracy ($\overline{d_b}$) was 0.2±2.94°C (range: −6.3 to 8.9°C; Fig. S4A) and absolute value ($|\overline{d_b}|$) was 2.4±1.71°C (range: 0–8.9°C; Fig. S4B); the minimum value was for an individual in water and the maximum value was for an individual undergoing ecdysis. Daily maximum $T_b$ ($VT_{max}$) across surveys was 30.1±2.67°C. Notable remarks for $VT_{max}$ are that max temperature occurred during afternoon scans for eight of ten survey days, for hidden (i.e. unexposed) individuals seven of ten times, and twice for individuals undergoing ecdysis.

Four competing LMM models (<2 ΔAICc) best described $T_b$ of *T. rufipunctatus* (Table S3). The best performing model included $T_{perch}$, $T_s$-$T_a$, exposed, and shared microhabitat behavior (AICc=391.3, weight=0.184, $R^2$=0.83); only the latter term failed to meet the importance threshold (>0.8). Competing models with additional terms had relative importance well below the 0.8 threshold and their inclusion failed to improve AICc >2. For each unit increase in $T_{perch}$ or $T_s$-$T_a$ in the optimal model, $T_b$ increased by 0.87°C [confidence intervals (c.i.): 0.74–1.01°C; Fig. 3A] and decreased by 0.32°C (c.i.: 0.14–0.51°C; Fig. 3B), respectively, after accounting for other terms. A categorical shift from hidden to exposed behavior (i.e. surface-visible) yielded a 0.42°C decline in $T_b$ (c.i.: 0.08–0.76°C), holding other terms constant.

### Behavioral assessment

We observed actively moving individuals only five times (4.0%; Table 2). We observed only four instances of activity in water but note that the complexity of the aquatic environment and natural behavior of the species could have obscured detection. Gartersnakes were partially or entirely exposed – to the visual scan of the observer – in 21 of 124 detections (16.9%). We encountered more exposed gartersnakes during morning surveys (23.7%) than afternoon surveys (6.3%, $\chi^2$=5.2, $P$=0.023; Table 2). Hidden snakes were warmer (27.5±2.72°C) than exposed individuals (25.7±3.60°C, $P$=0.045; Fig. 4), suggestive of afternoon retreat

into refuges once upper bounds of preferred temperature were obtained from morning warm up. Gartersnakes were warmer in the afternoons (29.4±2.18°C) than mornings (25.8±2.51°C, $P$<0.001) but there was no interactive effect of exposure ($P$=0.123).

Exposed behavior was best explained by $T_a$ and $T_s$ ex-in (AICc=46.9, weight=0.461, $R^2$=0.78; Table S4). For each unit increase in $T_a$, the odds of shifting between hidden to exposed behavior decreased by 0.33 times (c.i.: 0.11–0.92), i.e. −67%; gartersnakes were most likely to be hidden when late-morning, early-afternoon air temperature reached ca. 28°C (Fig. S5). There were some trends that probability of exposure increased with greater differences between external and internal microhabitat surface temperature but with uncertainty (confidence intervals crossed 1).

Gartersnake aggregations were observed 31 times (39.7% of occurrences). The mean number of aggregated snakes was 2.5 (±0.77). The maximum number of snakes sharing a given microhabitat was four, which was recorded five times; most aggregations occurred within refuges such as small hide boxes or hibernacula. Aggregations occurred more often during the morning (64.5%) than the afternoon (35.5%; Table 2), and aggregations per month from May through September (2019) were three, six, four, nine, and nine, respectively. There were no significant differences in $T_b$ for gartersnakes in aggregations versus solitary ones ($F_{114}$=2.6, $P$=0.111).

Four competing GLMMs best explained aggregation behavior at microhabitats (Table S5). All contained average $T_b$ from present individuals, which was the only factor meeting the importance threshold. The top performing model included average $T_b$ and $T_{perch}$ (AICc=96.1, weight=0.200, $R^2$=0.48). After accounting for other terms, each unit increase in average $T_b$ decreased the probability of aggregation by 0.52 times (c.i.: 0.30–0.91), i.e. 48% decline: warmer gartersnakes were less likely to aggregate (Fig. 5). There were trends that increasing $T_{perch}$ led to higher aggregation probability but with uncertainty (confidence intervals crossed 1). Three competing models best explained aggregation quantity, each contained average $T_b$ from present individuals and/or $T_{perch}$ (Table S5). Neither of these factors, however, met the variable importance threshold (<0.8) and did not appear to influence aggregations due to uncertainty about the

**Table 1. Comparative body temperatures (°C) of *T. rufipunctatus* during its active season (ca. March–November)**

| *n* | Mean (°C) | s.d. (°C) | Min.–max. (°C) | Method | Source |
|---|---|---|---|---|---|
| 31 | 24.7 | ±2.2* | 20.0–29.0 | QCT | Fleharty, 1967 |
| 18 | 25.0 | ±3.7 | 19.2–31.4 | QCT | Rosen, 1991 |
| 60 | 26.4 | ±2.7 | 19.8–30.4 | QCT | Hibbitts et al., 2009§ |
| 15 | 26.5 | ±4.3 | 17.3–32.0 | QCT | Jennings and Christman¶ |
| 8 | NA | NA | 10< $T_b$ <35‡ | TRT | Nowak¶ |
| 117 | 27.2 | ±3.0 | 18.5–33.7 | IRT | Present study |

Methods of assessment include cloacal temperature (QCT; quick-read thermometer), temperature-sensing radio transmitter (TRT; internally implanted), and infrared thermography (IRT; external/dermal). Sample size (*n*) may include repeated measurements across time. *=converted from standard error; ‡=inferred from graphical data (active season only); §= author provided additional data; ¶=unpublished report, see Holycross et al. (2020); USFWS (2014).

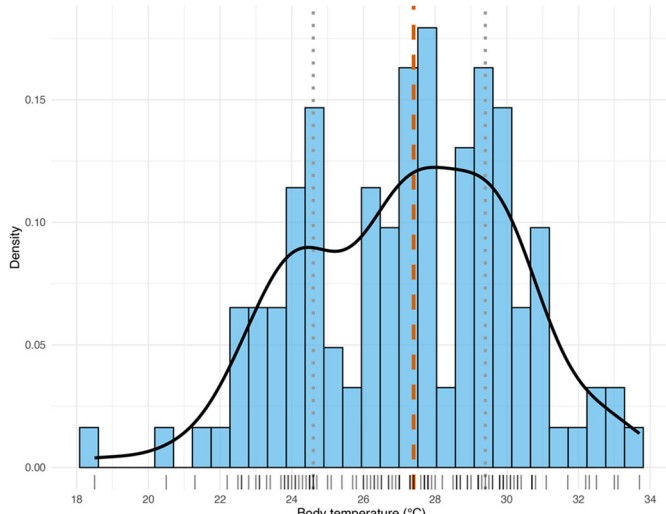

**Fig. 2. Distribution of body temperature (*n*=117) for *ex situ* colony of *T. rufipunctatus* at the Phoenix Zoo, 2019.** Vertical lines represent median (dashed, orange) and upper and lower quartile bounds of the preferred body temperature ($T_{set}$; dotted, grey). Tick marks along the x-axis represent individual datapoints.

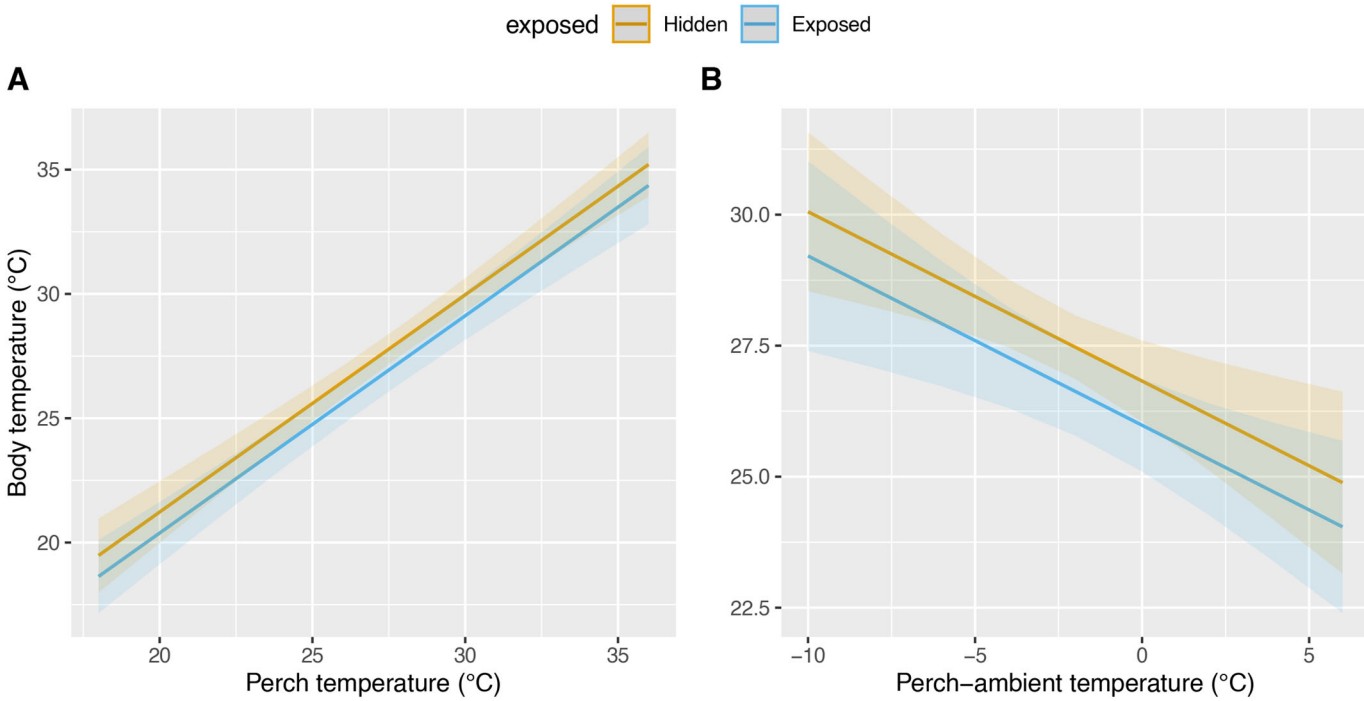

**Fig. 3. Predicted change in body temperature for *ex situ* colony of *T. rufipunctatus* at the Phoenix Zoo, 2019.** (A) Perch surface temperature and (B) the difference in perch temperature from ambient. Data are partitioned whether individuals were visibly exposed or hidden at a microhabitat.

confidence intervals (crossing 1). That is, the tested variables failed to explain an influence of aggregation quantity in the *T. rufipunctatus* colony.

## DISCUSSION
We used IRT, a reliable and noninvasive method, to infer surface body temperature for a colony of threatened *T. rufipunctatus* in naturalistic *ex situ* mesocosms. We examined extrinsic and intrinsic factors that influence microhabitat usage, $T_b$, and behavior. These insights into the seldom assessed thermal ecology and physiology of *T. rufipunctatus* serve as informational feedback loops to increase knowledge and steer conservation strategies across the *ex situ*–*in situ* spectrum. To our knowledge, this is the first application of IRT for *T. rufipunctatus* and on snakes in enriched mesocosms.

### Unraveling the thermal ecology of a threatened, limited-range ectotherm
Ecological insight into *T. rufipunctatus* ecology and life history is warranted (USFWS, 2014), and opportunities remain to investigate the physiology and thermal ecology for the species. The range of body temperatures of *T. rufipunctatus* from this study was slightly

warmer than reported from some wild populations (Fleharty, 1967; Hibbitts et al., 2009; Rosen, 1991). Snakes can maintain $T_b$ higher in lab settings than wild (Cox et al., 2023). However, the $T_b$ exhibited were equivalent to the most recent wild assessment (see Jennings and Christman in USFWS, 2014), which may better reflect contemporary climate shifts across the species' range (Blais and Koprowski, 2024; Pilliod et al., 2024). The ACNC colony's $T_b$ and $T_{set}$ were also within the range of a relatively high stable phase activity temperature exhibited by *T. rufipunctatus* and syntopic congenerics (Rosen, 1991). Several factors were influential in predicting $T_b$. The most prominent were perch temperature, the difference in microhabitat surface temperature from ambient, and behavior (particularly, exposure). As perch temperature rose, so did $T_b$ – concordant with findings in wild populations (Fleharty, 1967; USFWS, 2014) and other regional herpetofauna (Blais et al., 2023a; Dubiner et al., 2024). Body temperature also increased as ambient air temperature increasingly exceeded microhabitat surface temperature ($T_s$-$T_a$; see Fig. 3B). These emphasize heterogeneity of the mesocosm environment and allowed for behavioral shifts among microhabitats to occur.

Relatively few snakes have been documented to exhibit regional heterothermy (Cox et al., 2023, 2024). Our findings add

**Table 2. Frequency of behavior events by diel period for *ex situ* colony of *T. rufipunctatus* at the Phoenix Zoo, 2019**

| Disposition | Morning | | Afternoon | | Cumulative | |
|---|---|---|---|---|---|---|
| | Freq. (%) | Mean $T_b$ (±s.d.) | Freq. (%) | Mean $T_b$ (±s.d.) | Freq. (%) | Mean $T_b$ (±s.d.) |
| Active | 1 (1.4) | 28.0 (–) | 2 (4.4) | 30.0 (3.1) | 3 (2.6) | 29.3 (2.5) |
| Inactive | 71 (98.6) | 25.8 (2.5) | 43 (65.6) | 29.3 (2.2) | 114 (97.4) | 27.1 (2.9) |
| Exposed | 16 (22.2) | 24.8 (3.1) | 3 (6.7) | 30.4 (2.3) | 19 (16.2) | 25.7 (3.6) |
| Hidden | 56 (77.8) | 26.1 (2.3) | 42 (93.3) | 29.3 (2.2) | 98 (83.8) | 27.5 (2.7) |
| Solitary | 24 (33.3) | 26.4 (2.6) | 19 (43.2) | 29.5 (2.4) | 43 (37.1) | 27.7 (2.9) |
| Aggregated | 48 (66.7) | 25.6 (2.4) | 25 (56.8) | 29.3 (2.1) | 73 (62.9) | 26.8 (2.9) |

Disposition refers to gartersnake behaviors, recorded *ad libitum* during surveys: Active=actively moving or swimming; Inactive=immobile/stationary; Exposed=partially or fully surface-visible to observer; Hidden=unexposed; Solitary=1 gartersnake per microhabitat; Aggregated=≥2 gartersnake per microhabitat. Freq.=frequency of events (%); $T_b$=body temperature.

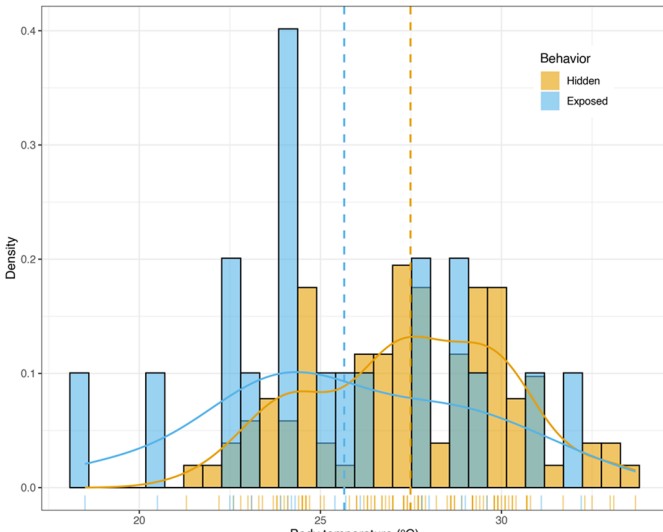

**Fig. 4. Behavioral differences in external body temperature (*n*=117) for *ex situ* colony of *T. rufipunctatus* at the Phoenix Zoo, 2019.** Data are partitioned by visually exposed (blue) versus hidden (orange) gartersnakes. Vertical dashed lines indicate sample means and tick marks along the x-axis represent individual datapoints.

*T. rufipunctatus* to this list with warmer head/trunk than tail patterns, similar to other temperate congenerics (Gregory, 1990). These differences were significant at about 0.4°C, which is less variable than observed in other species (e.g. 1.7–2.8°C in *Diadophis punctatus*; Cox et al., 2024). This may be due to adaptations as a semi-aquatic piscivore operating at cooler temperatures than congenerics (Fleharty, 1967; Rosen, 1991), and a prehensile tail important for underwater foraging strategies (e.g. anchoring in swift-currents; Holycross et al., 2020). Diel/seasonal timing, instrument type, and internal-specific measurement could also affect heterothermy inference. Because heterothermy can persist across behaviors (e.g. exposed versus hidden) and regardless of thermal environments – albeit at lesser extents as temperature increases,

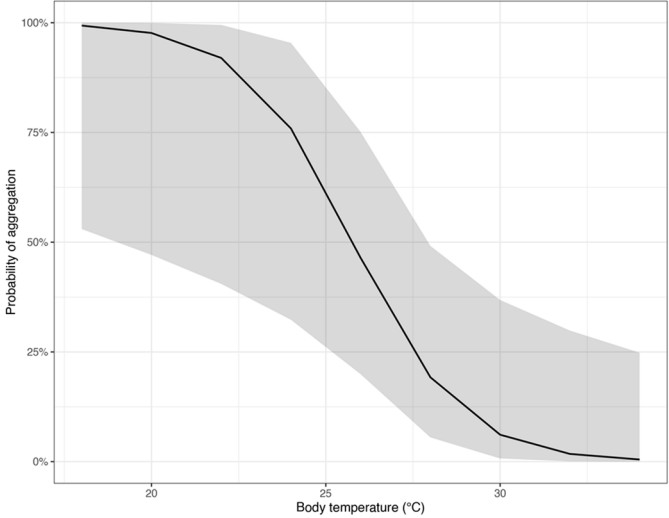

**Fig. 5. Probability of aggregation behavior based on body temperature for *ex situ* colony of *T. rufipunctatus* at the Phoenix Zoo, 2019.** Aggregation is defined as ≥2 snakes (versus solitary individuals) at a given microhabitat.

endogenous mechanisms may be at play in thermoregulating vital areas such as the head/brain (Cox et al., 2024; Tattersall et al., 2006). Regional heterothermy in *T. rufipunctatus* may have more to do with physiological necessities than environmental influence; further research on underlying functional mechanisms of regional heterothermy is warranted.

Exposed behavior also influenced $T_b$ but, interestingly, hidden snakes tended to be warmer than exposed snakes. This result is likely temporal. More frequent exposed individuals in the morning hours (cooler body temperatures, basking) versus warmer but more likely hidden gartersnakes in the afternoon could be explained by afternoon retreats into refuges when mean $T_b$ approximated the $T_{set}$ upper bounds; gartersnakes likely reached their upper preferentia sometime in the mid to late morning and had retreated into cover by time the early afternoon surveys occurred. This finding is supported by our assessment of exposed behavior, which was influenced by ambient air temperature; gartersnakes were more likely to be hidden once ambient temperature exceeded ca. 28°C across the survey season. These findings lend further support of a morning warmup and afternoon retreat in *T. rufipunctatus* (see literature in USFWS, 2014) and other snakes (Venegas-Barrera et al., 2025). Because visual observations were done *ad libitum*, we acknowledge uncertainty whether individuals had recently shifted from basking to cooling or vice versa.

Numerous factors can influence microhabitat selection by reptiles, including thermal and non-thermal properties (Blais et al., 2023a; Campobello et al., 2017; Mushinsky and McCoy, 2016; Ortega et al., 2019). *Thamnophis rufipunctatus* is often detected atop or within riparian rock piles, vegetation, and other structures with areas for quick escape (Hibbitts et al., 2009; Holycross et al., 2020; USFWS, 2014); they have also been observed occupying pipes and other manmade structures in parts of its range (see Nowak in USFWS, 2014). For the *ex situ* colony of *T. rufipunctatus* at the ACNC, we found that microhabitat type was the best predictor of selection, particularly cover objects that had internal cavities for refuge. Despite microclimate heterogeneity among the microcosms, no other factor appeared to influence microhabitat occupancy in this study; untested variables (e.g. dimensionality, solar radiation) may provide further resolution (Ortega et al., 2019).

Squamate reptiles, including *Thamnophis* gartersnakes, aggregate for a variety of reasons across inter/intraseasonal periods, such as overwinter denning, beneath cover during active seasons, gestation, and for mating (Aleksiuk, 1977; Gardner et al., 2016; Graves and Duvall, 1995; Gregory, 2004). Vomeronasal mechanisms guide individuals into aggregations (Aubret and Shine, 2009; Heller and Halpern, 1982). Aggregations in wild *T. rufipunctatus* populations are sparsely reported (Blais and Lashway, 2018; Holycross et al., 2020; USFWS, 2014). Our study is the first to examine links between thermal ecology and aggregation behavior in *T. rufipunctatus*. We identified a relationship between $T_b$ and whether individuals aggregated or not; warmer gartersnakes were less likely to (or continue to) aggregate. None of our tested variables, however, explained aggregation cohort size. Congenerics in temperate zones are known to aggregate to prevent water loss, albeit concentrated during overwinter denning (Costanzo, 1989), whereas during the active season, aggregations to reduce cooling rates tend to favor neonates/younger (i.e. smaller) snakes (see Aubret and Shine, 2009 for a review on squamate aggregation). A 'selfish herd' defense mechanisms for aggregation, especially during vulnerable periods such as female gestation (Gregory, 2016) could also exist; one female reproduced during our study year (Wood et al., 2025b). More recently, some *Thamnophis* were found to aggregate socially by engaging in

nonrandom social interactions (Skinner and Miller, 2020). Failing to find statistical support among our tested variables for aggregation cohort size lends to two basal explanations: 1) the available data were insufficient to test the variables and/or, 2) we failed to examine the variable(s) that explain clustering, such as social reasons (Skinner and Miller, 2020). Our noninvasive 'no touch' design prohibited closer examination for sex and individual ID, though we recommend that future investigations apply mark-recapture techniques (e.g. PIT tags, nontoxic paint markings) to track any social, reproductive, or rate-of-change evidence therein. In the wild, observers should take note of microhabitat type/structure, sex, size, gravidity, and developmental age class of aggregating individuals.

### Implications across the *ex situ*–*in situ* spectrum

There were several indicators that the *T. rufipunctatus* in the ACNC colony are exhibiting behavioral and physiological traits akin to those observed in wild populations. Body temperatures were equivalent to the most recent known assessment in the wild (see Jennings and Christman in USFWS, 2014), and thermoregulatory accuracy was comparable to typical values in other squamates (Hertz et al., 1993; Ivey et al., 2020). Moreover, the lack of month significantly influencing $T_b$ infers that *T. rufipunctatus* were able to thermoregulate across the active season (Ivey et al., 2020). Taken together, this suggests there was sufficient microclimate heterogeneity for the colony to adequately thermoregulate; individuals on the outer margins occurred during spring mornings or in water (cooler end) or were undergoing ecdysis in summer afternoons (warmer end). Behaviorally, gartersnakes were seldom observed moving and usually were well hidden and sometimes undetectable to the brief visual scan despite the finite environment. Similar traits have been observed in the wild (Holycross et al., 2020; Rosen, 1991; USFWS, 2014). Enriched *ex situ* populations managed in semi-natural environments can exhibit innate behaviors and physiology (Blais et al., 2022; Burghardt, 2013; Rabier et al., 2022), which may be key for producing fit individuals to maximize opportunity for *in situ* conservation success (Blais et al., 2025; Choquette et al., 2023; Gross et al., 2024; Péchy et al., 2015; Roe et al., 2015). Collectively, these findings support that the *ex situ* colony is engaging in direct measures of natural defense and survival behaviors (Chiszar et al., 1993) while helping build the ecological knowledge base for *T. rufipunctatus* (Blais and Lashway, 2018; Holycross et al., 2020).

When *ex situ* populations exhibit normal traits and processes, the inferences obtained from empirical research may better reflect conditions expected in the wild and guide more informed field sampling and monitoring strategies (Blais et al., 2023b; Chiszar et al., 1993; Roe et al., 2015). Locating the rare and declining *T. rufipunctatus* in the wild can be challenging (Ryan et al., 2019; USFWS, 2014). Deciphering the relationships between environmental temperatures and both body temperature and behavior may hint at when and where to find snakes (Davis et al., 2008). We provide further evidence (Holycross et al., 2020; USFWS, 2014), for example, that surveys earlier in the day ($T_a$ <28°C) during the late spring through summer months may be optimal for detecting basking individuals, whereas searches beneath cover may be more fruitful during mid-day heat when individuals have likely retreated to cool off.

### Conservation benefits and future directions

*Thamnophis rufipunctatus* have declined throughout their range – attributed to wildfire, habitat degradation/loss, and competition/predation from non-native invasive species (Holycross et al., 2020; USFWS, 2014) – culminating in small, genetically isolated, and vulnerable populations (Wood et al., 2018, 2025a). Distributions

correlate with overwinter temperature and dry season precipitation (Blais and Koprowski, 2024). Concerningly, their occupied region is expected to get warmer and drier (Archer and Predick, 2008; Cook et al., 2015; Georgescu et al., 2022), with substantial reduction in suitable environmental range projected under future emissions scenarios (Blais and Koprowski, 2024). Conservation recommendations call for further examining ecology and life history, managing habitats for occupancy, and increasing genetic diversity and viability of populations through translocations, including individuals from conservation breeding programs (Holycross et al., 2020; USFWS, 2014; Wood et al., 2025a). The insights from this study mark a step in understanding the thermal ecology of this riparian-dependent, cool-tolerant ectotherm in a range that is likely to be adversely affected by climate change (Blais and Koprowski, 2024).

The critical next steps include assessing thermal maxima and adaptive potential in *T. rufipunctatus* and other ectotherms under threat of climate change (Bodensteiner et al., 2021; Griffis-Kyle et al., 2018; Hof et al., 2011; Kearney et al., 2009; Monge et al., 2025). Advances in IRT technology enable accurate and real-time monitoring of body and environmental temperature across thermal landscapes (Blais et al., 2023a). Paired with remote cameras, IRT may reveal diel, seasonal, and long-term spatiotemporal thermographic trends *in situ* and *ex situ* (Alujević et al., 2025; Mochales-Riaño et al., 2024; Rowe et al., 2020). Multidisciplinary studies integrating IRT and genomics could diagnose evolutionary potential of thermal tolerance traits (Monge et al., 2025). Spatially broader IRT scans may help detect 'climate refugia' (Keppel et al., 2015; Scheffers et al., 2014; Suggitt et al., 2018) critical for future management (Blais and Koprowski, 2024; USFWS, 2021). Resolution into other dynamics of behavior and thermal ecology by developmental age class – such as diel/nocturnal activity, brumation hibernacula, and gestation sites – would be informative (Aleksiuk, 1977; Aubret and Shine, 2009; Blais et al., 2023a; Krochmal and Bakken, 2003; Reiserer et al., 2008; Signore et al., 2020).

This study and others (Fleharty, 1967; Rosen, 1991) shows that *T. rufipunctatus* depends on microhabitat structure, indicating the importance of thermal heterogeneity. For management purposes, maintaining and restoring microhabitat structure and complexity across riparian and upland buffer zones is likely paramount (Scheffers et al., 2014; Suggitt et al., 2018; Woods et al., 2015); artificial cover (e.g. concrete boards) may provide substitutes when natural refuges are lost (Lelièvre et al., 2010). Potential collaborations among wildlife managers and zoos/aquariums should promote naturalistic environments in conservation breeding programs to promote essential behaviors (e.g. sociality, reproductive needs, foraging strategies, etc.; Blais et al., 2022; Eisenberg and Kleiman, 1977; Minteer et al., 2018). Doing so may present optimal arenas for *ex situ* studies benefiting *in situ* conservation and translocations (Blais et al., 2025; Gross et al., 2024; Spooner et al., 2023). *Ex situ* managers of *T. rufipunctatus* or other terrestrial ectotherms should monitor responses and maintain key microhabitat conditions, e.g. hides/burrows, water temperature (Hibbitts et al., 2009; Holycross et al., 2020; Murphy et al., 1994; Rosen, 1991). A deeper exploration into intrinsic, extrinsic, and spatiotemporal drivers of thermal ecology, occupancy, and microhabitat selection will refine species and habitat conservation strategies.

## MATERIALS AND METHODS

### System and experimental design

The ACNC's narrow-headed gartersnake enclosures and husbandry management have been adaptively retrofitted to incorporate naturalistic

conditions aiming to optimize snake welfare and stimulate reproductive activity (Allard and Wells, 2018). Gartersnakes were managed communally for sociality and pedigree purposes (Blais et al., 2022; Wood et al., 2025b) in two (4.88×2.44×2.44 m) adjacent climate-controlled walled enclosures with a translucent sheet roof that allowed for natural solar photoperiodism. Each enclosure had a dug-out and lined pond with circulated and aerated water and a small waterfall to simulate natural plunge-pool stream habitat. Live fish were managed in the aquatic zones for natural foraging. Each enclosure had a sunken and climate-controlled hibernaculum chamber for brumation; further details of enclosure design, environmental parameters, and cohort origins/pedigree are described elsewhere (Blais et al., 2022, 2023b; Wood et al., 2025b).

We enumerated microhabitats available across both enclosures (*n*=42) and categorized their type as cover (e.g. bark slabs, plastic zoological hide boxes, PVC half-pipes), ground (e.g. single/stacked rock slabs, open areas), plants (e.g. leafy vegetation, branches/sticks), subterranean (i.e. hibernacula and its cover slab), and water (e.g. pool with submerged and partially-emergent river rocks). The program's design aimed to simulate environmental heterogeneity and, because cohabitation and free-movement was unimpeded (i.e. the snakes choose when/where to move, feed, and engage in brumation), innate behaviors are stimulated, including aquatic-adjacent basking, courtship, and reproduction (Blais et al., 2022, 2023b). The enclosures thus act as mesocosms comparative to natural systems used by *T. rufipunctatus*.

We conducted biweekly IRT surveys, May–September 2019, coinciding within the narrow-headed gartersnake active season and especially during the gestation/parturition period (Blais et al., 2023b; Goldberg, 2003; Holycross et al., 2020). We performed separate late morning and early afternoon survey shifts in each enclosure (i.e. four surveys/date) to encompass diel activity trends (Holycross et al., 2020); snakes can exhibit daily thermoregulatory patterns that include a morning ramp up and a mid-day stabilization phase (Venegas-Barrera et al., 2025). At the beginning of each survey, we used a handheld anemometer (Kestrel 5000, Nielsen-Kellerman Co., Boothwyn, Pennsylvania, USA) to measure ambient conditions at 1 m including air temperature ($T_a$, ±0.1°C), relative humidity (rH, ±0.1%), and barometric pressure ($P_b$, ±0.1 millibars). Water surface temperature ($T_w$, ±0.1°C; HI98129, Hanna Instruments, Smithfield, Rhode Island, USA) was recorded daily by ACNC staff. Because certain environmental stimuli may influence gartersnake behavior and body temperature (Rosen, 1991; Row and Blouin-Demers, 2006; Venegas-Barrera et al., 2025), we categorically estimated cloud cover and recent rainfall (scored binomially if occurred ≤48 h); we obtained precipitation data from a nearby weather station (<5 km, Tempe, AZ, USA). Photoperiod and seasonal climate patterns (e.g. summer monsoonal onset) in Phoenix, Arizona, are comparable to natural populations circa ±2° latitude away (Blais et al., 2023b). We preliminarily used Wilcoxon rank sum tests to compare ambient conditions ($T_a$, rH, $P_b$; nonnormally distributed) and $T_w$ between enclosures. Because none of the tested conditions differed between enclosures (Wilcoxon tests: *P*>0.05), we used a combined dataset for analyses.

To derive surface temperature ($T_s$) of microhabitats, we used a FLIR E8 infrared thermal camera (FLIR Systems, Wilsonville, OR, USA) to capture multispectral thermograph/photograph combinations [resolution=320×240 (76,800) pixels, accuracy ±2%; thermal sensitivity <0.06°C] at a height approximately 0.5–1 m (Blais et al., 2023a). We ensured emissivity was set to 0.97, commonly used with reptiles (Barroso et al., 2016; Blais et al., 2023a; Luna and Font, 2013). For microhabitats with both external and internal surfaces (e.g. bark slabs, hide boxes), we first scanned the exterior surface before gently lifting the object to scan the underlying substrate surface; this external-internal process may better encompass the microclimate heterogeneity available to snakes (Cox et al., 2023).

When gartersnakes were visibly present on or within microhabitats, we captured additional images that included full body length (dorsal side) whenever possible (see Fig. S1). Dermal surface temperature of relatively small squamates captured by IRT, for example, is correlated with cloacal or internal body temperature (Barroso et al., 2016; Rowe et al., 2020; Tattersall, 2016), which, for snakes, is related to environmental temperature (Cox et al., 2023, 2024). We scored microhabitat occupancy (snake absence, presence) and quantified aggregations (i.e. *n*≥2; Gregory, 2004). To minimize disturbance, only a single observer (B.R.B.) performed surveys,

and gartersnakes were not handled due to noninvasive design. Inherently, some individuals became alerted to observer presence, including when refuge covers were temporarily lifted for scans. We assigned binary scores to the following behaviors: 'exposed' (i.e. partial or entire body exposed to surface elements and visible to observer upon first detection=1; or hidden entirely within a microhabitat=0); 'moving' (if a gartersnake was actively locomoting across terrestrial or aquatic zones=1, immobile=0); and 'shared' (if a given microhabitat was shared with more than one individual=1, or not i.e. solitary=0). The number of gartersnakes present in each enclosure was known prior to each survey (colony maximum *N*=11 mature individuals) but not all individuals were detected each survey; we only analyze data on detected snakes. Sexes within enclosures were mixed for pedigree management (Wood et al., 2025b), but individual ID or sex could not be discerned due to our noninvasive design. Body size is not known to influence external body temperature in snakes (Cox et al., 2023; Venegas-Barrera et al., 2025) but can sometimes vary by sex (Rosen, 1991; Venegas-Barrera et al., 2025). This research was conducted under board approval from the Arizona Center for Nature Conservation/Phoenix Zoo's Conservation Department; the ACNC is accredited by the Association of Zoos and Aquariums (AZA) and meets AZA standards of animal care and welfare.

## Thermographic assessment

We used diagnostic tools in FLIR Tools software v. 6.4 (FLIR Systems) to generate descriptive statistics (±0.1°C), where each pixel in a thermograph represented a datapoint. Parallel to Blais et al. (2023a). We used polygon tool functions in attempt to capture the maximum coverage of microhabitat $T_s$ without overlapping onto adjacent microhabitats or snake body surfaces. Because surface water temperature measurements were equivalent between IRT and ACNC's water meter ($t_{17}$=0.4, *P*=0.69), we retained only the IRT data hereafter for consistency.

We used a combination of spot, line, and polygon functions in FLIR Tools (Blais et al., 2023a) to generate $T_b$ (±0.1°C) of gartersnakes. We assessed regional heterothermy (i.e. $T_b$ differences by body segments; Sannolo et al., 2014; Barroso et al., 2016, 2020) by investigating dorsal integuments separately as follows: head measurements included the center of the skull and from the snout to back of head when possible; trunk (mid-body) measurements included linear segments (lines, polygons) from behind the head to approximately the cloaca (Cox et al., 2023) – ca. 0.67 of the body length; and tail measurements (when observable) occurred along the posterior third of individuals, where tail width tapers after the cloaca. If an integument had >1 measurement, we estimated averages; we omitted an individual's integument sections if it was not visibly discernable from thermograph/photograph combinations. In preliminary testing, we found no differences between line or polygon functions which suggests either can be used advantageously pending the positioning of subjects. Because this study uses IRT to gain insights into the thermal ecology and physiology of a threatened ectotherm in a naturalistic zoological setting, we were more interested in comparative relationships of body temperature and environment – more applicable for conservation practitioners – rather than finite calibrations of internal $T_b$ (Blais et al., 2023a; Mazzamuto et al., 2023; Playà-Montmany and Tattersall, 2021).

## Statistical analyses

Throughout and where applicable, we used Shapiro-Wilk tests and QQ-plots to assess data normality and a Spearman correlation matrix to ensure highly correlated variables (|*r*|>0.7) were not combined in downstream analyses. We used Wilcoxon tests and Kruskal–Wallis test to assess differences in ambient conditions ($T_a$, rH, $P_b$, $T_w$) by diel shift and month, respectively. To examine environmental and temporal influences of microhabitat occupancy (used=1, available=0; Objective 1), we used mixed effect logistic regression (GLMM) from the R package lme4 (Bates et al., 2015) against the following suite of predictors: mean microhabitat surface temperature ($T_s$), the difference in mean $T_s$ to ambient air temperature ($T_s$-$T_a$), relative humidity (rH), microhabitat type, diel shift, and month. We set microhabitat ID as a random effect. We used a function in the glmulti package (Calcagno, 2020) to exhaustively automate the top performing models within two corrected AICc units from the global lme4 model candidates. We note that manually performing backwards selection in lme4 yielded the same model conclusions as those automated in glmulti, supporting the latter's utility for swift and reliable output. To refine

further, we assessed relative importance values for predictors where values ≥0.8 signaled cumulative weighted importance in determining a top model (Calcagno and de Mazancourt, 2010). Because there was an apparent preference towards microhabitats with internal structures, e.g. hide boxes, bark slabs, and hibernacula, we created a separate dataset for only microhabitats with internal components and repeated the above steps. For this subset, we removed covariates $T_s$ (only accounted for external surfaces) and $T_s$-$T_a$ and added $T_a$, $T_s$ in (internal microhabitat substrate temperature), and $T_s$ ex-in (the difference in microhabitat external surface versus internal substrate temperature).

We derived several metrics to understand body temperature and thermal physiology of *T. rufipunctatus* (Objective 2). We first tested for regional heterothermy of integuments with a two-way type III repeated-measures analysis of variance (rmANOVA) with Greenhouse-Geisser corrections for sphericity via the afex package (Singmann et al., 2024); we only included individuals that had head, trunk, and tail measurements. We included exposure (i.e. exposed versus hidden) as a grouping factor, which may help account for potential heating versus cooling behaviors. We used averaged $T_b$ values derived from thermographs for each integument, which can better account for heterothermy (Barroso et al., 2016; Blais et al., 2023a; Rowe et al., 2020).

We estimated total mean $T_b$ per individual by averaging integument temperatures; at least one of head or trunk integuments was always used. We performed single-sample Z tests to compare the sample population mean $T_b$ in this study to estimates (μ, s.d.) obtained from wild *T. rufipunctatus* populations (Fleharty, 1967; Hibbitts et al., 2009; Rosen, 1991; USFWS, 2014). We considered preferred body temperature ($T_{set}$) as the median $T_b$ value and the $T_{set-range}$ as the first and last quartiles, i.e. interquartile range (Blouin-Demers and Weatherhead, 2001; Hertz et al., 1993; Taylor et al., 2021). We estimated thermoregulatory accuracy ($\overline{d_b}$) in two ways: first, we generated the absolute value difference between the average median value (i.e. $T_{set}$) from mean $T_b$ (Hertz et al., 1993; Taylor et al., 2021); we also derived the raw difference in $T_{set}$-$T_b$, which may better estimate poor thermoregulation at preferred warmer temperatures ($d_b < 0$) or poor cooling to the preferred body temperature ($d_b > 0$; Ivey et al., 2020; Taylor et al., 2021). We conservatively inferred $VT_{max}$ (i.e. voluntary maximum temperature before animal retreats to shelter to avoid further heating; Taylor et al., 2021) by averaging the maximum $T_b$ per survey date (Ivey et al., 2020); we acknowledge these estimates are relative and may not encompass the full suite of thermal physiology and confounding effects experienced in a further-controlled thermal arena or natural setting (Terblanche et al., 2011).

To understand environmental factors that may influence $T_b$ in *T. rufipunctatus* (Objective 3), we followed Blais et al. (2023a) by conducting a candidate suite of general linear mixed models (LMM) with Gaussian distribution. Predictors included occupied (surface) perch temperature ($T_{perch}$), $T_s$-$T_a$, rH, $P_b$, exposed, shared, and month; we set surveys per shift (morning versus afternoon) as random effects. Because $T_w$ was correlated with both $T_{perch}$ and $T_a$, and gartersnakes were seldom observed in water, we did not include it as a model parameter. We omitted standalone $T_a$ as a predictor due to high correlation with, and lower performance (e.g. AICc, $R^2$) than $T_{perch}$; perch temperature has shown to be a stronger predictor of $T_b$ than $T_a$ for other regional herpetofauna (Blais et al., 2023a). We again used glmulti, AICc, and predictor relative importance as described above to select an optimal model among candidates. We used sjPlot (Lüdecke, 2024) to predict and plot values from resulting models.

To examine factors that may influence exposed (surface-visible) behavior – which may help guide field surveillance – we assessed logistic GLMM candidates in glmulti with the following environmental predictors: $P_b$, rH, $T_a$, $T_s$ ex-in, and shift; we set month and survey as random effects. Finally, for aggregations ($n \geq 2$, Gregory, 2004) in *T. rufipunctatus* (Objective 4), we conducted a series of GLMMs in glmulti in two parts. First, we asked what influences aggregations at microhabitats (binary: solitary versus aggregations ≥2)? Next, we asked what influences aggregation quantity, where the response variables were counts of individuals present at a given microhabitat? We used logistic GLMM for the former and Poisson GLMM for the latter. In both cases, predictors included $T_{perch}$, $T_s$-$T_a$, rH, shift, month, and the average mean $T_b$ from all individuals present; the latter aims to explain intrinsic influences. We set individual microhabitat ID and

surveys as random effects. We repeated the model assessment workflow as described above. Data are presented as the mean±s.d. unless stated otherwise. We report Nakagawa & Schielzeth's $R^2$ for optimal models. We performed all analyses in program R (R Core Team, 2021).

**Acknowledgements**
We would like to thank staff and personnel from the Arthur L. and Elaine V. Johnson Conservation Center at the Arizona Center for Nature Conservation/Phoenix Zoo, especially B. Poynter, T. Harris, and R. Allard, for permitting the study and providing enclosure access, data, and other collaborative insight. We are grateful to S. Wells for helpful comments about the colony design for *T. rufipunctatus*, T. Hibbitts for providing comparative data, and S. Johnson for data processing assistance.

**Competing interests**
The authors declare no competing or financial interests.

**Author contributions**
Conceptualization: B.R.B., M.V.M., J.L.K.; Data curation: B.R.B.; Formal analysis: B.R.B.; Investigation: B.R.B.; Methodology: B.R.B., M.V.M.; Project administration: B.R.B., J.L.K.; Resources: J.L.K.; Software: B.R.B.; Supervision: J.L.K.; Visualization: B.R.B.; Writing – original draft: B.R.B.; Writing – review & editing: M.V.M., J.L.K.

**Funding**
 Deposited in PMC for immediate release.

**Data and resource availability**
All relevant data and details of resources can be found within the article, its supplementary information, and from Figshare: https://figshare.com/s/184f5e70c996751e2346.

**Peer review history**
The peer review history is available online at https://journals.biologists.com/bio/lookup/doi/10.1242/bio.062264.reviewer-comments.pdf

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
