## [Peer Review File · Biology Open]

Insights into the thermal ecology, physiology, and behavior of a threatened ectothermic specialist from a warming and drying ecoregion

Maria Vittoria Mazzamuto, John L. Koprowski and Brian R. Blais

DOI: 10.1242/bio.062264

Editor: Lewis Halsey

Review timeline

Original submission: 17 September 2025

Editorial decision: 22 September 2025

First revision received: 28 November 2025

Accepted: 2 December 2025

Original submission

First decision letter

MS ID#: bio.062264

MS Title: Insights into the thermal ecology, physiology, and behavior of a threatened ectothermic specialist from a warming and drying ecoregion

Authors: Brian R. Blais, Maria Vittoria Mazzamuto and John L. Koprowski

I have now reached a decision on the above manuscript.

The reviewer reports are shown at the bottom of this email or can be accessed, together with a copy of this decision letter, by going to:

As you will see, the reviewers gave favourable reports, but raised some critical points that will require amendments to your manuscript. I hope that you will be able to carry these out, because we would like to be able to accept your paper.

At this stage, we also ask you to ensure your manuscript complies with our formatting guidelines - please see our manuscript preparation guidelines for details. Provided you are able to fully address the referees' comments, we are positive about publication of your paper (we accept over 95% of revision submissions) and therefore hope you won't mind any extra work involved in reformatting your manuscript at this point.

Please upload both a 'clean' version of your Word file, along with a highlighted version clearly showing where you have made changes in the revised manuscript. Please avoid using 'Track changes' in Word files as these are lost in PDF conversion.

I should be grateful if you would also provide a point-by-point response detailing how you have dealt with the points raised by the reviewers in the 'Response to Reviewers' box. Please attend to all of the reviewers' comments. If you do not agree with any of their criticisms or suggestions please explain clearly why this is so.

Reviewer 1

Comments for the author

The study describes body temperature measurements of an endangered species of snakes in enclosures that mimic their natural habitat. The results are analyzed at depth, and I get the impression of a thorough study and I have only a few remarks.

Materials and Methods:

When describing the enclosures please state early on that they were outdoors. This only became clear to me further down when rainfall was mentioned. Also, what refers to side lengths and height in the measures given for the enclosures (line 131)?

I found no description of the genetic background of the snakes studied. Are they the offspring of a few or many individuals? How long has the colony existed - several generations? Parameters like temperature preferences are likely to have a genetic background.

Line 170: It is unclear to me what "+/- 2 °C; sensitivity <0.06 °C" means. The sensitivity of the thermal camera could be better described. What was the accuracy/precision in °C? Did it need regular calibration and was this done? That could be important over such a long observation period (May - September).

Figures:

What does the small vertical lines on the x-axis in Fig 2, 4 and S4 show? Individual observations? Please explain in legends.

Discussion:

It was found that cold snakes were more likely to aggregate. I found this quite interesting and the reason for aggregations are discussed. Could one additional reason be that they try to retain metabolic body heat by reducing the collective skin surface area exposed? Some homeotherms are known to do this and theoretically ectotherms could also benefit from it even if their metabolic heat production is much less, but not insignificant (about 1/10th). Has it ever been suggested for snakes?

Reviewer 2

Comments for the author

In their study, Blais et al. used non-invasive thermal imaging to measure and explore the interplay between thermoregulatory and behavioural responses of a rare garter snake to various microhabitats. The authors study an ex-situ population of animals in zoos or similar conditions to avoid the natural stressors present in the wild but to still recapitulate the natural habitat and thermal physiology of this threatened species. This approach could also allow future studies to consider life-history and health-status, although this didn't seem to be the case in the present study design. In the present study, the authors aimed to shed light on thermal and habitat preferences of this species to inform future field surveillance and population management strategies. Overall, the paper is very clearly written and provides sufficient background and review of the field to allow the reader to understand the importance of their contribution. The methods (and especially the statistical analysis) and results are clearly presented with a comprehensive set of data provided. The discussion touches on the major observations of the study, compares the results to previous work to validate both the method and the biological observations, and offers considerable insight into how the results can be used to inform future conservation and population management strategies in a warming world. There are some limitations to the study, mostly pertaining to the use of a "point in time" strategy that assayed animal behaviour in two specific windows, missing animals that were not detected in each survey, and limited pictures available to assay Tb relative to total

observations of animals detected. However, the authors do a good job of explaining their experimental design choices and the value of the data presented is not overly diminished by these limitations. I am generally enthusiastic about this submission and have only a few relatively minor comments for the authors to consider.

Minor Comments:

1. Is more information available about the interrelation within the animals under study? I am curious to learn if these animals were related (from a common source within the zoo's breeding program) or represent genetically distinct animals from various wild-caught populations. If they are all from a common source then the authors should consider/discuss the impact of such limitations on their experimental subject pool as they may be colony effects .

2. Measurements were only taken ~ mid-day (but in two shifts). This thus represents a dataset for a particular and small window in the behaviour of this species, which would likely correlate with a warm part of the day. It would have been interesting to have measurements across 24 h but I understand that this may not have been the goal of the others and would be challenging to achieve. Nonetheless, a bit more emphasis on the limitations of this sampling window should be addressed in the Discussion.

3. The authors indicate they conducted 18 surveys per enclosure across a period of 4 months (~ 18 weeks) but then say the average survey interval was 14 days...this doesn't make sense. Please check this math and revise.

4. Lines 340-343 - I appreciate that the authors have provided a statistical comparison of their data to previously published work and the mean and error from these previous studies. Some information of the study/field site temperature between these studies would also be useful, given the generally ectothermic nature of snakes. A brief discussion of the impact of your measurement approach (method) relative to the previous studies and how this may impact the accuracy of a given measurement would also be useful.

5. The regional heterothermy in this study is a very small difference (0.4C). Given that the authors admit they were not concerned with exact Tb measurements, the use of a surface-level means to measure Tb, inexact borders for delineating anatomical thermal regions in a picture, etc., I wonder if this difference is real and/or biologically significant. I wonder if the authors could expand on some of these limitations a bit more, given the relatively rarity of regional heterothermy in snakes, which makes their claim important if accurate.

6. The authors take a common Tb by averaging temps across the three body regions but the body region makes up most of the total body length. The impact of the small tail region on total Tb is thus amplified by the authors taking an average of the 3 zones without weighting the various variables by % of body length. Some correction for this might impact the significance of the Tb difference between visible and exposed animals, etc.

Reviewer's Responses to Questions

Experimental quality

Does each figure have the proper controls?

If 'No', please indicate reasons in Comments for Author box below.

Reviewer #1:

- Yes

Reviewer #2:

- Yes

Were the data analyzed using appropriate statistical tests?

If 'No', please indicate reasons in Comments for Author box below.

Reviewer #1:

- Yes

Reviewer #2:

- Yes

Reproducibility

Were experiments performed using adequate number of biological replicates?

If 'No', please indicate reasons in Comments for Author box below.

Reviewer #1:

- Yes

Reviewer #2:

- Yes

Does the methods section provide sufficient detail to permit reproducibility?

If 'No', please indicate reasons in Comments for Author box below.

Reviewer #1:

- Yes

Reviewer #2:

- Yes

Completeness

Are the manuscript's conclusions supported by the data?

If 'No', please indicate reasons in Comments for Author box below.

Reviewer #1:

- Yes

Reviewer #2:

- Yes

Scholarship

Do the authors cite and discuss the merits of data that would argue for and against their conclusion?

If 'No', please indicate reasons in Comments for Author box below.

Reviewer #1:

- Yes

Reviewer #2:

- Yes

Does the manuscript title & abstract accurately reflect the contents of the manuscript, without hyperbole?

If 'No', please indicate reasons in Comments for Author box below.

Reviewer #1:

- Yes

Reviewer #2:

- Yes

First revision

Author response to reviewers' comments

Comments from the Reviewers:

Reviewer 1: The study describes body temperature measurements of an endangered species of snakes in enclosures that mimic their natural habitat. The results are analyzed at depth, and I get the impression of a thorough study and I have only a few remarks.

Materials and Methods:

When describing the enclosures please state early on that they were outdoors. This only became clear to me further down when rainfall was mentioned. Also, what refers to side lengths and height in the measures given for the enclosures (line 131)?

We revised to clarify that enclosures were walled with a translucent overhead (see lines 127-128). This design excludes rain/wind but allows exposure to certain conditions (e.g., solar, atmospheric pressure). Although rain would not penetrate into enclosures, for example, changes in pressure would be sensed and cloudiness could influence lighting/brightness. The onset of seasonal precipitation (and associated factors) can trigger life history events for some snakes in the region (in lit) which justified our initial inclusion of that data as an event (binary). To remain succinct, we refer readers to another study that provides more details about the enclosure design (Blais et al. 2022; line 131-133).

I found no description of the genetic background of the snakes studied. Are they the offspring of a few or many individuals? How long has the colony existed - several generations? Parameters like temperature preferences are likely to have a genetic background.

We noted that the ex situ program began in 2006 (line 100; see Blais et al 2022 for more detail on program origin), and that pedigree information is detailed elsewhere (lines 131-133; Blais et al., 2022; Wood et al., 2025b). Although genetics/genomics is outside the scope of this study, we do cite all relevant genetic works on *T. rufipunctatus* - primarily conducted by Wood and others (2011, 2018, 2025a/b). We added “genetically” to clarify our use of isolated in “...culminating in small, **genetically** isolated, and vulnerable populations...” (Lines 524-525). Also in the Discussion, we suggest that IRT + genetics could offer resolution into evolutionary potential of focal species (lines 542-544). To our knowledge, there are no known intraspecific genetic signatures associated with thermal preferences for *T. rufipunctatus*.

Line 170: It is unclear to me what “ $\pm 2^\circ\text{C}$; sensitivity $<0.06^\circ\text{C}$ ” means. The sensitivity of the thermal camera could be better described. What was the accuracy/precision in $^\circ\text{C}$? Did it need regular calibration and was this done? That could be important over such a long observation period (May - September).

This was helpful for us to clarify the specs of the E8 camera, including resolution (320 x 240 pixels), accuracy (2% [not $^\circ\text{C}$...a typo] and *thermal* sensitivity ($<0.06^\circ\text{C}$). FLIR thermal cameras come pre-calibrated (with much detail in their user manual, which need not use up words in the manuscript) and we verified settings prior to each use. We adjusted the language around these parameters/methods for clarity (lines 164-169).

Figures:

© 2025. Published by The Company of Biologists under the terms of the Creative Commons Attribution License (<https://creativecommons.org/licenses/by/4.0/>).

What does the small vertical lines on the x-axis in Fig 2, 4 and S4 show? Individual observations? Please explain in legends.

To all captions in question, we added “...Tick marks along the x-axis represent individual datapoints.”

Discussion:

It was found that cold snakes were more likely to aggregate. I found this quite interesting and the reason for aggregations are discussed. Could one additional reason be that they try to retain metabolic body heat by reducing the collective skin surface area exposed? Some homeotherms are known to do this and theoretically ectotherms could also benefit from it even if their metabolic heat production is much less, but not insignificant (about 1/10th). Has it ever been suggested for snakes?

An interesting thought. Some lizards will splay out to increase surface contact for conductive heating (increases) or reduce the amount of surface contact (e.g., limbs elevated off contact) if trying to cool down - tubular snakes don't tend to have such a luxury and are more likely to shift to cover once sufficient heating (or reaching maxima) occurs. Some thermal advantages of aggregating are discussed but an understanding of reasoning still warrants further investigation, especially as most snakes don't aggregate altogether and those that do tend to be during communal overwintering or during youth (smaller size)...albeit contemporary research is finding less asociality than previously thought. It's all very fascinating, but our design did not provide the kinds of data needed to test such things. To that end, we added statements to better paint the additional plausibility in the Discussion (lines 473-477; and in parallel response to Reviewer 2).

Reviewer 2: Overall comments:

In their study, Blais et al. used non-invasive thermal imaging to measure and explore the interplay between thermoregulatory and behavioural responses of a rare garter snake to various microhabitats. The authors study an ex-situ population of animals in zoos or similar conditions to avoid the natural stressors present in the wild but to still recapitulate the natural habitat and thermal physiology of this threatened species. This approach could also allow future studies to consider life-history and health-status, although this didn't seem to be the case in the present study design. In the present study, the authors aimed to shed light on thermal and habitat preferences of this species to inform future field surveillance and population management strategies. Overall, the paper is very clearly written and provides sufficient background and review of the field to allow the reader to understand the importance of their contribution. The methods (and especially the statistical analysis) and results are clearly presented with a comprehensive set of data provided. The discussion touches on the major observations of the study, compares the results to previous work to validate both the method and the biological observations, and offers considerable insight into how the results can be used to inform future conservation and population management strategies in a warming world. There are some limitations to the study, mostly pertaining to the use of a "point in time" strategy that assayed animal behaviour in two specific windows, missing animals that were not detected in each survey, and limited pictures available to assay Tb relative to total observations of animals detected. However, the authors do a good job of explaining their experimental design choices and the value of the data presented is not overly diminished by these limitations. I am generally enthusiastic about this submission and have only a few relatively minor comments for the authors to consider.

Minor Comments:

1. Is more information available about the interrelation within the animals under study? I am curious to learn if these animals were related (from a common source within the zoo's breeding program) or represent genetically distinct animals from various wild-caught populations. If they are

all from a common source then the authors should consider/discuss the impact of such limitations on their experimental subject pool as their may be colony effects .

We essentially addressed a similar comment from another reviewer and redirect to that response above. In brief, pedigree is covered elsewhere (Blais et al., 2022; Blais et al., 2023b; Wood et al., 2025b; lines 131-133), and individuals did not originate from a common source. We also suggest broader applications of IRT and in combination of other resources which would add further resolution (lines 536-549).

2. Measurements were only taken ~ mid-day (but in two shifts). This thus represents a dataset for a particular and small window in the behaviour of this species, which would likely correlate with a warm part of the day. It would have been interesting to have measurements across 24 h but I understand that this may not have been the goal of the others and would be challenging to achieve. Nonetheless, a bit more emphasis on the limitations of this sampling window should be addressed in the Discussion.

We surveyed during known peaks of activity (late morning + early afternoon) for *T. rufipunctatus* (and many snakes) based off the lit and personal expertise (lines 144-149, 448-450). In the Discussion, we additionally revised to add "...diel, seasonal, and long-term ..." to the statement "Paired with remote cameras, IRT may reveal diel, seasonal, and long-term spatiotemporal thermographic trends *in situ* and *ex situ* (Alujević et al., 2025; Mochales-Riaño et al., 2024; Rowe et al., 2020)" (lines 540-542). We also added "...diel/..." to the statement "Resolution into other dynamics of behavior and thermal ecology by developmental age class—such as diel/nocturnal activity, brumation hibernacula, and gestation sites—would be informative" (lines 546-549). These should give readers a sense that added temporal coverage may lead to further insights.

3. The authors indicate they conducted 18 surveys per enclosure across a period of 4 months (~ 18 weeks) but then say the average survey interval was 14 days...this doesn't make sense. Please check this math and revise.

We clarified that our biweekly design of two surveys per enclosure (n=2) per sampling date (line 147), resulted in 36 total surveys (1-2 per enclosure per sampling date) after accounting for two afternoon shifts omitted due to logistics (lines 296, 300-301).

4. Lines 340-343 - I appreciate that the authors have provided a statistical comparison of their data to previously published work and the mean and error from these previous studies. Some information of the study/field site temperature between these studies would also be useful, given the generally ectothermic nature of snakes. A brief discussion of the impact of your measurement approach (method) relative to the previous studies and how this may impact the accuracy of a given measurement would also be useful.

This was helpful as we forgot to cite our Table 1 in the lines mentioned - it contains the data and collection methods used in the comparative studies. Sufficient detail of Tb relationships to ground or air temperature (outside of regression equations or graphics) or data subsets of exposed vs hidden snakes are lacking in the comparative studies, but because of the relatively restricted distribution of cool-stream specialist *T. rufipunctatus*, we would expect environmental compositions among field sites to not vary to the (spatial) extent it would compound results. As ectotherms, snake body temperature is relatable to their environment, whether in nature or a naturalistic mesocosm, and prior evidence supports IRT as a reliable metric for inferring body temperature (see lines 65-71). In the Discussion (lines 409-417), we refer to contemporary and forecasted increases in temperature for a focal region (Blais and Koprowski 2024) compared to earlier comparative works—i.e., environmental temperatures now and into the next century are certain to be warmer than those gathered in the 20th ce. but we offer evidence-supported assessment (Tb ~ Tg + Ta) via technique (IRT) that allows for close monitoring across time. We capture these ideas in our Introduction (effectiveness of IRT), experimental design, but especially in the *Conservation/Future Directions* section of our Discussion (lines 522-535). We also draw

attention to revised future directions statements where IRT (and in combination with other technologies) could provide further resolution (lines 536-549).

5. The regional heterothermy in this study is a very small difference (0.4C). Given that the authors admit they were not concerned with exact Tb measurements, the use of a surface-level means to measure Tb, inexact borders for delineating anatomical thermal regions in a picture, etc., I wonder if this difference is real and/or biologically significant. I wonder if the authors could expand on some of these limitations a bit more, given the relatively rarity of regional heterothermy in snakes, which makes their claim important if accurate.

The mechanisms that facilitate regional heterothermy in snakes are poorly known, but not seemingly (unanimously) maintained through behavioral thermoregulation alone (see Cox et al 2024; Tattersall et al., 2006). While our design/scope omits exploration into the mechanism(s) behind regional heterothermy, we expanded on our Discussion context to allude to other physiological means for such and heterothermy in general (lines 426-439).

6. The authors take a common Tb by averaging temps across the three body regions but the body region makes up most of the total body length. The impact of the small tail region on total Tb is thus amplified by the authors taking an average of the 3 zones without weighting the various variables by % of body length. Some correction for this might impact the significance of the Tb difference between visible and exposed animals, etc.

We can appreciate this concern but reiterate how data were captured. Unlike spot-type laser thermometers that capture only a single reading per scan, we opportunistically used the FLIR E8 thermal camera's multi-datapoint captures (thermogram res 320 x 240 pixels, thermal sensitivity <0.06C) among body integument. We clarified that "...each pixel represented a datapoint" (Lines 198-201) and by using separate multi-pixel line/polygon values for the head, body (trunk), and tail integuments (see line 207), we derived a cumulative mean of those values. From these, we estimated a *total body temperature*, i.e., a robust amount of data for the estimation [see also Barroso et al., 2016; Blais et al., 2023a; Rowe et al., 2020 for how averaged data better accounts for heterothermy].

Posterior TB in snakes can be cooler than anterior regions for a variety of reasons (e.g., diel/seasonal environmental conditions, physiological function, etc. see Cox et al., 2024, Tattersall et al. 2006, and our response to #5 above). We remain confident that our systematic capture of multi-pixel thermal data across snake bodies adequately and sufficiently approximates total body TB specific to our questions within our scope. ...We also had no means to reliably infer % length/coverage by integument as our noninvasive design (no handling/moving snakes) prevented us from measuring body lengths by individuals, and forcing a generic % assumption would be too speculative, as % integument area varies by sex (Tanner 1990; Holycross et al 2020) and undoubtedly by developmental age class. In any case, we re-ran analysis using each bodily integument (head, trunk, tail) separately under the assumption that differing results would mean body region has a confounding effect. Because conclusions were unchanged to those with *total* (multi-integument) TB, we retain our original methods, results, and conclusions. We added a statement that individual integument assessments did not alter results (line 335). We believe our additional context on heterothermy (see #5 above) also addresses this inquiry.

Second decision letter

MS ID#: bio.062264R1

MS Title: Insights into the thermal ecology, physiology, and behavior of a threatened ectothermic specialist from a warming and drying ecoregion

Authors: Brian R. Blais, Maria Vittoria Mazzamuto and John L. Koprowski

I've had the opportunity this morning to read through your rebuttal and the associated edits to your manuscript. I consider your responses to be thorough and compelling, and therefore I am happy to tell you that your manuscript has been accepted for publication in Biology Open, pending our standard publication integrity checks. It was accepted on 2nd December 2025.